# Changes in Eating Behaviors and Their Associations with Weight Loss in Japanese Patients Who Underwent Laparoscopic Sleeve Gastrectomy

**DOI:** 10.3390/nu15020353

**Published:** 2023-01-10

**Authors:** Yu Kimura, Yuya Fujishima, Hitoshi Nishizawa, Takuro Saito, Yasuhiro Miyazaki, Keiko Shirahase, Chie Tokuzawa, Naoko Nagai, Shiro Fukuda, Kazuhisa Maeda, Norikazu Maeda, Yuichiro Doki, Iichiro Shimomura

**Affiliations:** 1Department of Metabolic Medicine, Graduate School of Medicine, Osaka University, Osaka 565-0871, Japan; 2Department of Gastroenterological Surgery, Graduate School of Medicine, Osaka University, Osaka 565-0871, Japan; 3Department of Surgery, Osaka General Medical Center, Osaka 558-8558, Japan; 4Division of Nutrition Management, Osaka University Hospital, Osaka 565-0871, Japan; 5Department of Metabolism and Atherosclerosis, Graduate School of Medicine Osaka University, Osaka 565-0871, Japan

**Keywords:** eating behavior, metabolic and bariatric surgery, obesity, emotional eating, weight loss

## Abstract

Background: Metabolic and bariatric surgery (MBS) has been established to provide long-term weight loss in severe obesity. In this study, we investigated the factors that affect post-operative weight loss, with a particular focus on changes in eating behaviors. Methods: Time-course changes in body weight and eating behaviors were examined in 49 Japanese patients who underwent laparoscopic sleeve gastrectomy from the first visit to 12 months after surgery. Each eating behavior was evaluated via the questionnaire of the Japan Society for the Study of Obesity. Results: Pre-operative weight reduction mediated by dietary and lifestyle interventions showed significant positive correlations with weight loss outcomes at 12 months after surgery. We observed significant decreases in scores for most of the eating behaviors 12 months after surgery. However, “emotional eating behavior” scores declined temporarily in the early post-operative period of one month but thereafter returned to the pre-operative level at 12 months. Furthermore, increases in the scores for “emotional eating behavior” and “sense of hunger” from 1 to 12 months post-operatively were significantly associated with poor weight loss. Conclusions: Our results demonstrate the beneficial effects of MBS on obesity-related eating behaviors, as well as highlighting “emotional eating behavior” as requiring particular attention.

## 1. Introduction

The number of patients with obesity and type 2 diabetes is increasing worldwide, especially in East Asian countries, including Japan [1,2]. As a result, metabolic and bariatric surgery (MBS) for patients with severe obesity has been increasing in recent years. In addition, its benefits for weight loss and obesity-related comorbidities, including diabetes mellitus, have been established [3,4,5]. However, the weight loss effect varies widely in individual cases, and post-surgical problems with psychiatric disorders, such as anxiety and depression, have been pointed out [6].

Obesity is caused by a combination of behavioral, psychosocial, and environmental factors, such as overnutrition, unhealthy eating habits, and physical inactivity, as well as genetic predispositions. As is especially the case with respect to severe obesity, genetic factors are indicated to play a relatively large part in its etiology [7]. Moreover, there are also a considerable number of cases of dysregulated eating behavior that are related to underlying psychological problems [8,9]. Thus, MBS requires careful consideration of the individual patient’s background. Furthermore, it is important not only to establish a long-term post-operative follow-up system but also to provide optimal nutritional and behavioral management during the pre-operative period. In this regard, morbidly obese patients often display various types of maladaptive eating behaviors, including feeding style, hunger sensations, food preferences, and emotional and external eating [10,11], which can interfere with weight loss interventions. However, the detailed effects of MBS on these obesity-related eating behaviors, as well as their relationships with weight loss outcomes, are still unclear, especially in Japanese patients.

In this study, we investigated the factors that affect weight loss outcomes after MBS throughout the pre- and post-operative periods from the first visit, with a particular focus on changes in eating behaviors.

## 2. Materials and Methods

### 2.1. Study Subjects

The present retrospective study was conducted by enrolling patients with severe obesity who underwent laparoscopic sleeve gastrectomy (LSG) at Osaka University Hospital between February 2014 and March 2021. The study protocol for this process is shown in Figure 1. We provided weight loss interventions consisting of medical treatment, individualized dietary counseling, and behavior modification prior to surgery, aiming at approximately 5% weight loss in conjunction with a comprehensive risk/benefit assessment. LSG was performed on 57 of the 88 patients who visited our hospital as candidates for MBS. The main reasons for not proceeding to surgery were as follows: Poor self-management capacity, self-withdrawal from surgery, the presence of uncontrolled psychiatric disorders, and interruption of hospital visits. Next, subjects who underwent LSG were followed up until 12 months after surgery. In addition, they received dietary counseling provided by registered dietitians for weight reduction and preventing weight regain at every visit. Their body weight was measured at the first visit, before surgery, as well as 1, 6, and 12 months after surgery.

### 2.2. Assessment of Eating Behaviors

Eating behaviors were assessed at the first visit, before surgery, and 1 and 12 months after surgery (Figure 1). These behaviors were evaluated using the questionnaire of the Japan Society for the Study of Obesity [10,11,12,13,14]. This method specifically identifies the problems in various eating behaviors of obese patients. Moreover, the questions are based on the words that are used by obese patients. This questionnaire comprises 55 questions on seven major scales as follows: (1) Recognition for weight and constitution (e.g., “Do you think it is easier for you to gain weight than others?”); (2) External eating behavior (e.g., “If food smells and looks good, do you eat more than usual?”); (3) Emotional eating behavior (e.g., “Do you have the desire to eat when you are irritated?”); (4) Sense of hunger (e.g., “Do you get irritated when you feel hungry?”): (5) Eating style (e.g., “Do you eat fast?”); (6) Food preference (e.g., “Do you like meat?”); (7) Regularity of eating habits (e.g., “Is your dinner time too late at night?”). All items are rated on a scale from 1(seldom) to 4(very often). By scoring the answers to the questionnaire according to gender and by constructing a diagram to confirm their characteristics, we can objectively grasp the problems of the patient’s eating behavior and eating habits. As the highest score differs between males and females for certain major scales, including “recognition of weight and constitution”, “external eating behavior”, “sense of hunger”, and “food preference”, we calculated the percentage to the full score on each eating behavior in order to quantitatively evaluate longitudinal changes after MBS in both genders.

### 2.3. Definitions

Type 2 diabetes was defined according to the criteria outlined by the World Health Organization (WHO) National Diabetic Group 2006 and/or according to the treatment of diabetes. Hypertension was defined as the use of anti-hypertensive drugs, systolic blood pressure at ≥140 mmHg, and/or diastolic blood pressure at ≥90 mmHg. Dyslipidemia was defined as the use of anti-hyperlipidemic drugs, low-density lipoprotein cholesterol (LDL-C) concentrations at ≥140 mg/dL, triglyceride (TG) concentrations at ≥150 mg/dL, and/or high-density lipoprotein cholesterol (HDL-C) concentrations at <40 mg/dL.

The percent of total weight loss (%TWL) and the percent of excess weight loss (%EWL) at 1, 6, and 12 months after surgery were calculated as follows: %TWL = (weight lost)/(body weight before surgery) × 100% and %EWL = (weight lost)/(body weight before surgery − ideal body weight (IBW)) × 100%. The IBW was defined as the body weight equivalent to a body mass index (BMI) of 22 kg/m^2^, according to Japanese criteria.

### 2.4. Statistical Analysis

All values are presented as the means ± standard deviation (SDs), medians (interquartile ranges (IQRs)), or the number of subjects (%). The correlations between total and excess weight loss and other clinical parameters were analyzed using Pearson’s correlation coefficient and scatter plots. Multiple regression analysis was conducted using age, sex, and pre-operative BMI as the covariates. Intergroup comparisons were tested with Student’s *t*-tests. The time-course changes in body weight, total and excess weight loss, and scores for each eating behavior were analyzed by repeated measures ANOVA followed by Tukey’s HSD tests. In all cases, a two-sided test was used, and *p* values <0.05 were considered statistically significant. All analyses were performed with JMP Statistical Discovery Software 15.0 (SAS Institute, Cary, NC, USA).

## 3. Results

### 3.1. Characteristics of the Study Subjects and Changes in Body Weight after Metabolic and Bariatric Surgery

LSG was performed on 57 of the 88 patients who visited our hospital as candidates for MBS. Of these, eight patients were lost to follow-up during the study, and thus the remaining 49 patients (25 males/24 females) were analyzed up to 12 months after surgery (Figure 1). The clinical characteristics of the study subjects are shown in Table 1. Briefly, the mean age and BMI at the first visit were 44.7 ± 8.7 years and 42.0 ± 8.3 kg/m^2^, respectively. Regarding obesity-related disorders, type 2 diabetes, hypertension, and dyslipidemia were observed in 55.1%, 57.1%, and 53.1% of the subjects, respectively.

Due to the medical management and repeated nutritional counseling for the purposes of pre-operative weight loss, it took 147 days (IQR = 104–242 days) from the first visit to the eventual surgery (Table 1). We observed a significant weight loss of 6.2% (from 118.0 ± 25.5 to 110.7 ± 21.5 kg) during the pre-operative period (Figure 2A). Then, the mean BMI and excess weight (BMI ≥ 22) before surgery was 39.8 ± 7.8 kg/m^2^ and 49.1 ± 19.8 kg, respectively (Table 1). Figure 2B,C show the time-course changes with respect to body weight, %TWL (Figure 2B), and %EWL (Figure 2C). LSG resulted in a significant continuous weight reduction until six months after surgery, and thereafter the rate of weight loss declined to a plateau level. The average %TWL and %EWL at 12 months after surgery was 21.3 ± 9.3% and 51.3 ± 25.8%, respectively, whereas relatively high variances with wide SDs (Figure 2B,C) indicated that some patients had insufficient weight loss.

### 3.2. Correlations between Pre-Operative Clinical Parameters and Weight Loss after Metabolic and Bariatric Surgery

First, we examined the relationships between each clinical parameter before surgery and weight loss outcomes at 12 months after surgery. When conducting a univariate analysis, pre-operative weight loss from the first visit to surgery showed significant positive correlations with both %TWL (*p* = 0.002) (Table 2 and Figure 3A) and %EWL (*p* = 0.001) (Table 3 and Figure 3B). These trends remained even after adjustment for sex, age, and pre-operative BMI (Std β = 0.44, *p* = 0.002 for %TWL; Std β = 0.53, *p* = 0.0007 for %EWL) (Table 2 and Table 3). Regarding obesity-related disorders, the presence of type 2 diabetes before surgery was associated with a significant decrease in %TWL in both the univariate and multivariate analyses (Table 2), as well as with a decreasing trend in %EWL (Table 3).

### 3.3. Changes in Eating Behavior Scores after Metabolic and Bariatric Surgery

Next, using the JASSO questionnaire, we accessed various types of eating behaviors before surgery as well as at 1 and 12 months after surgery. The scores for eating behaviors before surgery are shown in Table 1. Furthermore, their associations with %TWL and %EWL at 12 months after surgery are shown in Table 2 and Table 3, respectively. With regard to the univariate analysis, no significant correlation was found between the pre-operative scores for each eating behavior and either %TWL or %EWL. However, when adjusted for sex, age, and pre-operative BMI, patients with higher pre-operative scores for “recognition for weight and constitution” were associated with poorer %EWL (*p* = 0.017) (Table 3).

The time-course changes in each eating behavior after LSG are shown in Figure 4. Significant reductions in all of these eating behavior scores were observed one month after surgery. Although some of them, including “external eating behavior”, “emotional eating behavior”, “food preference”, and “regularity of eating habits”, increased from 1 to 12 months after surgery, all of the eating behavior scores, except for “emotional eating behavior”, decreased significantly by 12 months compared to those before surgery (Figure 4). On the contrary, the scores for “emotional eating behavior” declined temporarily in the early post-operative period of one month but then returned to the pre-operative levels by 12 months after surgery, with no significant change (Figure 4C).

### 3.4. Correlations between Changes in Eating Behavior Scores and Weight Loss after Metabolic and Bariatric Surgery

Finally, we examined the correlations between post-operative changes in each eating behavior score and weight loss outcomes at 12 months after surgery. No significant associations were found between either %TWL or %EWL and changes (Δ) in any eating behaviors from before to 12 months after surgery (Appendix A). On the other hand, as shown in Table 4, the change (Δ) in “emotional eating behavior” scores from 1 to 12 months after surgery showed a significant negative correlation with %TWL (*p* = 0.009). Furthermore, this relationship remained significant even following adjustment for sex, age, and pre-operative BMI (Std β = −0.37, *p* = 0.013). During this period, we also found significant negative correlations between %EWL and changes (Δ) in two eating behaviors, “emotional eating behavior” and “sense of hunger”, after conducting multivariate analysis (Std β = −0.36, *p* = 0.014 and Std β = −0.30, *p* = 0.045, respectively) (Table 5). 

## 4. Discussion

The major findings of this study were as follows: (1) Weight loss during the pre-operative period showed significant positive correlations with %TWL and %EWL at 12 months after surgery; (2) Eating behavior scores were significantly decreased by 12 months after surgery, except for “emotional eating behavior”, which declined temporarily in the early post-operative period but returned to the pre-operative levels at 12 months after surgery; (3) Increases in the scores for “emotional eating behavior” and “sense of hunger” from 1 to 12 months after surgery were significantly associated with lower weight loss outcomes.

To date, there have been several reports that have described the relationship between pre-operative weight loss and weight loss following MBS. Although these previous studies showed mixed results, the amount of weight reduction during the pre-operative period has been suggested as a potential positive predictor of one-year post-operative weight loss, as well as perioperative outcomes such as decreased operative time [15]. A recent systematic review, including 18 studies with at least two years of post-operative follow-up, reported a positive correlation between pre- and post-operative weight loss in four studies. However, it was found there was no correlation in 13 studies and a negative correlation in one study [16]. One possible reason for these inconsistent results across studies is the presence of methodological differences in weight loss programs or procedures used in the pre-operative period. Previous studies that reported negative or no correlations prescribed a very-low-calorie diet (VLCD) [17,18,19,20] or a liquid diet [21] for several weeks prior to surgery. On the contrary, a previous report, in which weight reduction before surgery was mediated primarily by relatively long-term (24–72 weeks) nutritional counseling, showed a significant positive correlation between pre-operative weight loss and %EWL at 12 months post-surgery [22]. Additionally, in another study, in which trained nurses introduced a 1200 kcal/day Mediterranean diet to the study subjects at least eight weeks prior to surgery, pre-operative weight loss was positively associated with %EWL at one and two years after surgery [23]. Likewise, the present study attempted weight reduction in LSG candidates through medical management and nutritional counseling without mandatory caloric restriction, thereby leading to a median pre-operative period of 147 days and an average weight loss of 6.2% before surgery. Additionally, we observed relatively strong positive correlations between pre-operative weight loss and both %TWL and %EWL at 12 months after surgery. These results, as well as ours, suggest that patients’ voluntary pre-operative weight loss through mid to long-term dietary and lifestyle modifications may affect the weight loss outcomes achieved by MBS. Collectively, pre-operative weight management based on closed medical and nutritional counseling is assumed to be an important step in predicting weight reduction after surgery.

Among the clinical parameters before surgery other than pre-operative weight loss, the presence of type 2 diabetes and high scores with respect to “recognition for weight and constitution” were associated with decreased %TWL and %EWL at 12 months after surgery. Previous studies have also demonstrated poor weight loss after Roux-en-Y gastric bypass (RYGB) in diabetic patients [24,25,26]. As insulin inhibits lipolysis and promotes lipogenesis and fat storage, the hyperinsulinemic conditions induced by insulin resistance and/or glucose-lowering medications with insulin or hypoglycemic agents that enhance endogenous insulin secretion are proposed as a potential cause of weight loss resistance in diabetic patients undergoing MBS [24,25,27]. High scores for “recognition for weight and constitution” imply a misunderstanding of one’s own obesity, as represented by the following questions: “Do you think it is easier for you to gain weight than others?”, “Do you think you gain weight even by drinking water?”, and “Do you think you cannot lose weight even though you do not eat so much?”. However, given the potential relationship with decreased weight loss after surgery, such pre-operative recognitional gaps regarding body weight may not always be derived from patients’ misperceptions. Genetic profiles have also been implicated in the etiology of obesity [7]. In addition, recent genome-wide association studies (GWAS) for BMI/obesity have identified more than 300 single-nucleotide polymorphisms (SNPs) [28]. Although it is still an area requiring active investigation, certain studies have addressed the poor response to surgical weight loss interventions in carriers of a high genetic risk score, calculated as the sum of adiposity-related SNPs [29,30,31]. Thus, high “recognition for weight and constitution” scores before surgery may, in part, reflect the genetic predisposition to obesity, thereby affecting weight loss outcomes after MBS.

The present study demonstrated, for the first time, the time-course changes in various types of eating behaviors in Japanese patients who underwent MBS. As a result, most of them, including in respect of “recognition for weight and constitution”, “external eating behavior”, “sense of hunger”, “eating style”, “food preference”, “regularity of eating habits”, and “the total score”, were significantly improved over the long post-operative period of one year. Regarding the correlations with weight loss outcomes, increased scores for “emotional eating behavior” and “sense of hunger” from 1 to 12 months after surgery were significantly associated with decreased %TWL and %EWL at 12 months (Table 4 and Table 5). With respect to this, gradual enlargement of the gastric volume following LSG after the subacute post-operative phase may explain a causal link between increased “sense of hunger” scores during this period and poor weight loss outcomes. When using the same questionnaire, we and others have shown the beneficial effects of glucagon-like peptide 1 receptor agonists (GLP-1 RAs), i.e., liraglutide and semaglutide, on body weight, as well as the multiple eating behaviors that were found in obese type 2 diabetic patients [10,14]. Moreover, in these previous studies, changes in “sense of hunger” scores were significantly correlated with changes in body weight after six months of treatment with GLP-1 RAs. Due to its peripheral and central actions of delaying gastric emptying and controlling hunger sensation [32], the enhanced GLP-1 secretion from the intestine is one of the mechanisms by which bariatric surgery procedures lead to massive weight loss [33,34]. In addition, lower increases in post-prandial GLP-1 levels have been reported to be associated with weight regain after RYGB [35]. Therefore, LSG-mediated alterations in the context of appetite-regulating gastrointestinal hormones, such as GLP-1, gastric inhibitory polypeptide (GIP), and ghrelin, may also be involved in the relationship between post-operative changes in “sense of hunger” and weight loss outcomes, but further research is needed to confirm this possibility.

Mental health problems with eating disorders, such as uncontrolled eating and binge eating, are very common among morbidly obese patients undergoing MBS [36,37]. This was demonstrated by the fact that there was a high frequency of binge eating disorders in 17% of MBS candidates [38]. Furthermore, in a meta-analysis, patients with binge eating disorders were suggested to have less post-operative weight loss and greater psychosocial distress after MBS [39]. Therefore, current guidelines worldwide [40], including in Japan [6], recommend the assessment of candidates for MBS in order to identify potential psychological factors that may compromise surgical outcomes. Emotional eating is an eating behavior that transfers psychological stress to food intake and has been indicated to reflect multiple negative emotions such as depression and anxiety, thereby leading to uncontrolled overeating [41]. Importantly, we found that, of the seven major scales of eating behaviors assessed by the questionnaire, only “emotional eating behavior” showed no significant change at 12 months after surgery, despite the temporal decline in the early post-operative period of one month (Figure 4C). Furthermore, elevation in “emotional eating behavior” scores from 1 to 12 months after surgery was significantly associated with decreased %TWL and %EWL at 12 months after surgery (Table 4 and Table 5). These results indicate that “emotional eating behavior” requires special attention, not only because it is difficult to improve in the long-term follow-up of MBS but also because its rebound increase after surgery interferes with weight loss outcomes. A previous study by Subramaniam et al. demonstrated that higher scores for emotional eating, as assessed by the Dutch eating behavior questionnaire (DEBQ), could be a negative predictor of %TWL six months after laparoscopic RYGB [42]. In the present study, scores for “emotional eating behavior” before surgery did not show any significant correlation with weight loss outcomes. However, we detected significantly higher “emotional eating behavior” scores at the first visit in the 31 patients who were not proceeded to surgery compared to those in the 57 patients who did undergo surgery (*p* = 0.026) (Appendix A and Appendix A). It is, therefore, possible that patients with obvious abnormalities in emotional eating were not included in this study. Taken together, repeated assessment of pre- and post-operative eating behaviors, including emotional eating, may be helpful for the purpose of better weight loss and its maintenance after MBS when combined with well-established weight management strategies for obesity, such as acceptance-based behavioral therapy [43].

There were several limitations in the present study. The limited sample size inhibited the ability to identify predictors of post-operative weight loss that possess smaller effects. Due to the fact that the assessment of each eating behavior was based on patients’ self-reporting, the risk of under or over-reporting cannot be ruled out. Additionally, we could not evaluate the frequency or quantity of food consumption; thus, it is still uncertain whether changes in eating behaviors were related to actual food intake. Since the single questionnaire was used in the present study, the results only assessed one aspect summarized in the questionnaire. Further prospective studies with larger sample sizes are needed in order to validate the finding obtained from the present study.

## 5. Conclusions

In summary, LSG for obese Japanese subjects significantly improved eating behaviors over the 12-month post-operative period, except in the case of “emotional eating behavior”. Moreover, weight loss outcomes at 12 months were associated positively with weight reduction during the pre-operative period and negatively with increased scores for “emotional eating behavior” and “sense of hunger” early after surgery. From our results, the questionnaire-based assessment of pre- and post-operative eating behaviors, especially with regard to “emotional eating behavior”, is proposed as a simple and useful method from which to provide appropriate dietary and behavioral interventions for the purpose of weight management in patients undergoing MBS.

## Figures and Tables

**Figure 1 nutrients-15-00353-f001:**
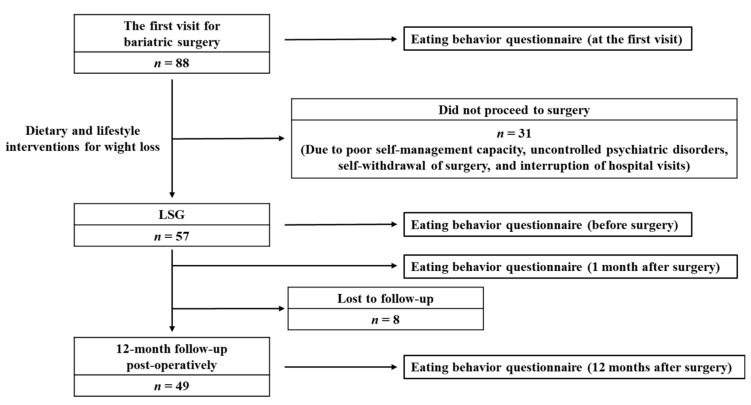
Enrollment and follow-up of the study subjects. Eighty-eight patients visited our hospital as possible candidates for metabolic and bariatric surgery. Of these, 57 patients who underwent laparoscopic sleeve gastrectomy (LSG) enrolled in the present study. Eight were lost to follow-up until 12 months after surgery and the remaining 49 patients were subjected to analysis.

**Figure 2 nutrients-15-00353-f002:**
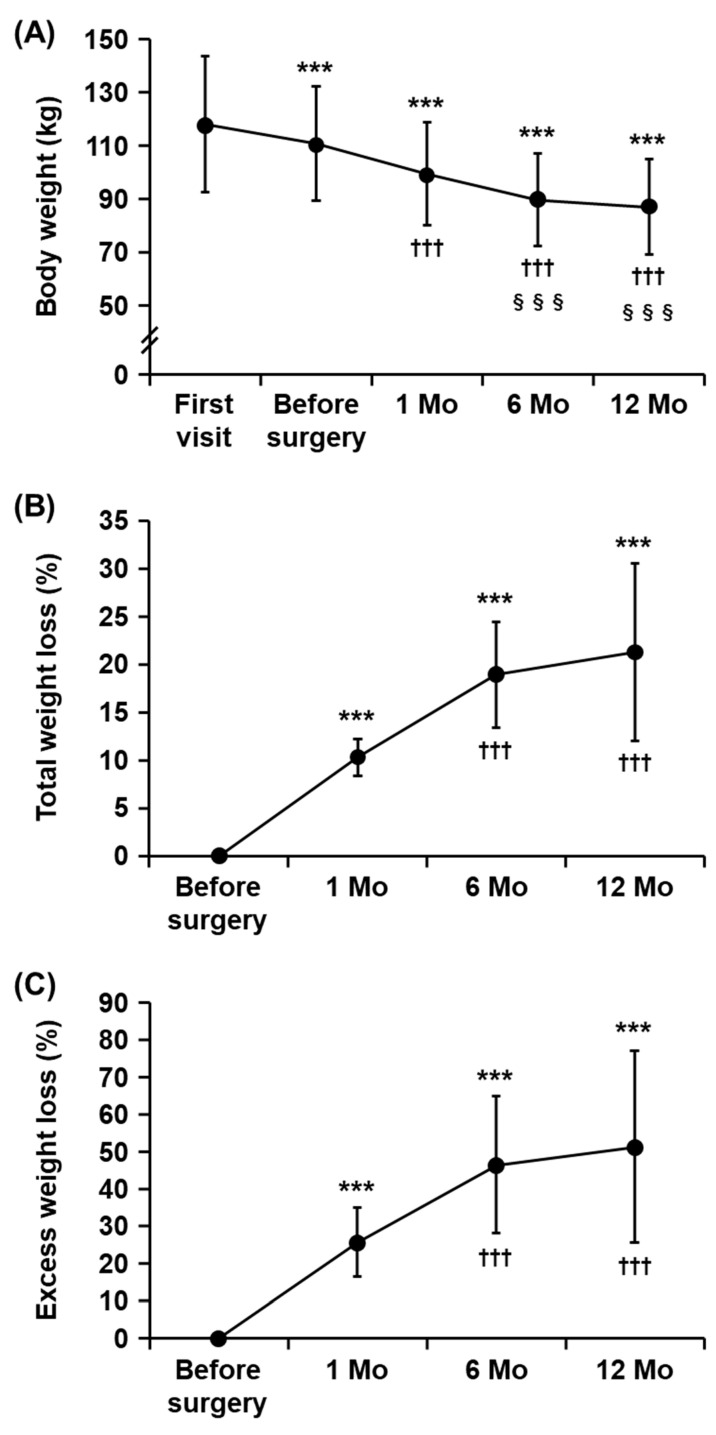
Time-course changes in body weight, total weight loss, and excess weight loss. (**A**) A change in body weight from the first visit to 12 months after surgery. *** *p* < 0.001 for the change from the first visit, ^†††^
*p* < 0.001 for the change from before surgery, and ^§§§^
*p* < 0.001 for the change from 1 month after surgery (repeated measures ANOVA with Tukey’s post hoc test). (**B**,**C**) The percent of total weight loss (**B**) and the percent of excess weight loss (**C**) at 1, 6 and 12 months after surgery. *** *p* < 0.001 for the change from before surgery and ^†††^
*p* < 0.001 for the change from 1 month after surgery (repeated measures ANOVA with Tukey’s post hoc test). Data are the mean ± SD. Mo, month.

**Figure 3 nutrients-15-00353-f003:**
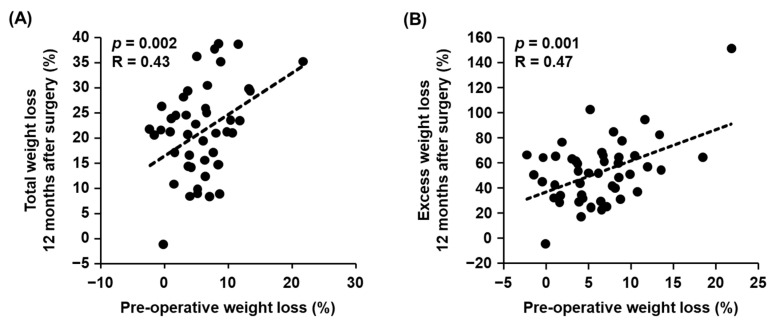
Relationship between weight loss during the pre-operative period and weight loss at 12 months after surgery. (**A**,**B**) Pearson’s correlation coefficient was used to examine the relationships between total weight loss during the pre-operative period and the percent of total weight loss (*p* = 0.002, R = 0.43) (**A**) and the percent of excess weight loss (*p* = 0.001, R = 0.47) (**B**).

**Figure 4 nutrients-15-00353-f004:**
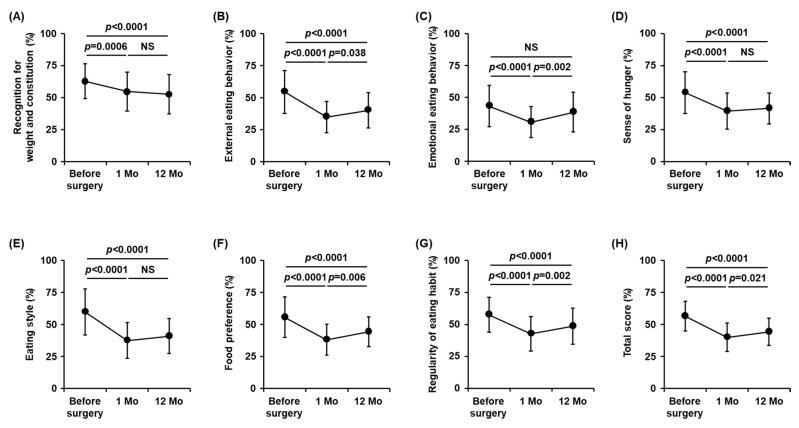
Time-course changes in scores for eating behaviors after surgery. (**A**–**H**) Changes in each eating behavior were assessed by the questionnaire, as described in the Section 2. Data are the mean ± SD. The *p* values were calculated using repeated measures ANOVA with Tukey’s post hoc test. Mo, month; NS, not significant.

**Table 1 nutrients-15-00353-t001:** Clinical characteristics of the study subjects.

Clinical Parameters	
*n* (males/females)	49 (25/24)
Age (years)	44.7 ± 8.7
Obesity-related complications	
Type 2 diabetes	27 (55.1%)
Hypertension	28 (57.1%)
Dyslipidemia	26 (53.1%)
(At the first visit)	
BW (kg)	118.0 ± 25.5
BMI (kg/m^2^)	42.0 ± 8.3
(Before surgery)	
BW (kg)	110.7 ± 21.5
BMI (kg/m^2^)	39.8 ± 7.8
Excess weight (kg)	49.1 ± 19.8
Eating behaviors	
Recognition for weight and constitution (%)	62.7 ± 13.6
External eating behavior (%)	54.5 ± 16.8
Emotional eating behavior (%)	43.2 ± 16.1
Sense of hunger (%)	54.0 ± 16.4
Eating style (%)	59.7 ± 18.0
Food preference (%)	55.5 ± 15.9
Regularity of eating habits (%)	57.6 ± 13.7
Total score (%)	56.5 ± 11.7
Days from the first visit to surgery	147 (104–242)

Data are presented as means (±SD), medians (IQRs), or the number of subjects (%). Abbreviations: BW, body weight; BMI, body mass index.

**Table 2 nutrients-15-00353-t002:** Correlations between clinical parameters before surgery and %TWL at 12 months after surgery.

			Sex, Age, and Pre-Operative BMI-Adjusted	

	Unadjusted			
Clinical Parameters before Surgery	R	*p* Value	Std β	*p* Value
Sex	-	0.560	-	-
Age	−0.19	0.199	-	-
BMI	0.18	0.208	-	-
Pre-operative weight loss (%)	0.43	0.002	0.44	0.002
Obesity-related complications				
Type 2 diabetes	-	0.006	0.35	0.020
Hypertension	-	0.420	0.05	0.764
Dyslipidemia	-	0.923	−0.03	0.860
Eating behaviors				
Recognition for weight and constitution	−0.18	0.222	−0.32	0.055
External eating behavior	0.05	0.753	0.03	0.858
Emotional eating behavior	−0.08	0.566	−0.14	0.347
Sense of hunger	0.10	0.519	0.05	0.768
Eating style	0.08	0.589	0.03	0.854
Food preference	−0.13	0.369	−0.23	0.131
Regularity of eating habits	0.04	0.773	0.005	0.974
Total score	0.01	0.968	−0.05	0.768

Abbreviations: %TWL, the percent total weight loss; BMI, body mass index.

**Table 3 nutrients-15-00353-t003:** Correlations between clinical parameters before surgery and %EWL at 12 months after surgery.

			Sex, Age, and Pre-Operative BMI-Adjusted	

	Unadjusted			
Clinical Parameters before Surgery	R	*p* Value	Std β	*p* Value
Sex	-	0.667	-	-
Age	−0.14	0.353	-	-
BMI	−0.28	0.054	-	-
Pre-operative weight loss (%)	0.47	0.001	0.53	0.0007
Obesity-related complications				
Type 2 diabetes	-	0.210	0.28	0.062
Hypertension	-	0.342	0.13	0.383
Dyslipidemia	-	0.742	−0.03	0.847
Eating behaviors				
Recognition for weight and constitution	−0.24	0.102	−0.38	0.017
External eating behavior	−0.12	0.426	−0.03	0.814
Emotional eating behavior	−0.23	0.119	−0.19	0.197
Sense of hunger	−0.09	0.534	−0.01	0.924
Eating style	0.03	0.854	0.04	0.794
Food preference	−0.22	0.127	−0.23	0.116
Regularity of eating habits	−0.05	0.713	−0.09	0.531
Total score	−0.19	0.190	−0.13	0.390

Abbreviations: %EWL, the percent excess weight loss; BMI, body mass index.

**Table 4 nutrients-15-00353-t004:** Correlations between changes in eating behavior scores from 1 to 12 months after surgery and %TWL at 12 months after surgery.

			Sex, Age, and Pre-Operative BMI-Adjusted	

Changes in Eating Behaviors (From 1 to 12 Months after Surgery)	Unadjusted			
	R	*p* Value	Std β	*p* Value
Δ Recognition for weight and constitution	−0.15	0.326	−0.13	0.419
Δ External eating behavior	−0.14	0.350	−0.16	0.308
Δ Emotional eating behavior	−0.38	0.009	−0.37	0.013
Δ Sense of hunger	−0.26	0.076	−0.29	0.060
Δ Eating style	−0.12	0.413	−0.17	0.281
Δ Food preference	−0.11	0.480	−0.06	0.710
Δ Regularity of eating habits	−0.17	0.263	−0.20	0.182
Δ Total score	−0.22	0.137	−0.24	0.127

Abbreviations: %TWL, the percent total weight loss; BMI, body mass index.

**Table 5 nutrients-15-00353-t005:** Correlations between changes in eating behavior scores from 1 to 12 months after surgery and %EWL at 12 months after surgery.

			Sex, Age, and Pre-Operative BMI-Adjusted	

Changes in Eating Behaviors (From 1 to 12 Months after Surgery)	Unadjusted			
	R	*p* Value	Std β	*p* Value
Δ Recognition for weight and constitution	−0.07	0.620	−0.12	0.437
Δ External eating behavior	−0.09	0.541	−0.12	0.432
Δ Emotional eating behavior	−0.27	0.062	−0.36	0.014
ΔSense of hunger	−0.29	0.048	−0.30	0.045
Δ Eating style	−0.11	0.458	−0.17	0.262
Δ Food preference	0.045	0.761	−0.08	0.616
Δ Regularity of eating habits	−0.19	0.192	−0.23	0.117
Δ Total score	−0.16	0.282	−0.24	0.115

Abbreviations: %EWL, the percent excess weight loss; BMI, body mass index.

## Data Availability

The data presented in this study are available on request from the corresponding author.

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
