# Peer review of "Changes in Eating Behaviors and Their Associations with Weight Loss in Japanese Patients Who Underwent Laparoscopic Sleeve Gastrectomy"

_nutrients, 2023, doi:10.3390/nu15020353_

Round 1

Reviewer 1 Report

Dear Authors,

I read the Manuscript. It is interesting, but it has many limits. It is retrospective and the number of patients is too small to be significative. The use of a single questionnaire cannot be easily comparable because it considers only the aspects summarized into it.

The follow up visits after surgery are a very small number and patients are not (or not described) helped to not regain weight.

I have just some suggestions for you:

1 - the title needs to be corrected. It is repeated twice.

2 - Figure 1: how many patients have been enrolled? 88 or 83? And how many patients underwent LSG? 57 or 55?

3 -Line 83: I think you could remove "as described previously" for what concerns  Japan Society for the Study of Obesity. You explain it in the following lines. 

Best regards

Author Response

Response to Reviewer #1

Reviewer comments for the author

I read the Manuscript. It is interesting, but it has many limits. It is retrospective and the number of patients is too small to be significative. The use of a single questionnaire cannot be easily comparable because it considers only the aspects summarized into it.

The follow up visits after surgery are a very small number and patients are not (or not described) helped to not regain weight.

Authors’ response

I greatly appreciate the reviewer’s important comments. According to these suggestions, the authors have described the study limitation of using the single questionnaire in the Discussion section, and nutritional guidance to prevent weight regain at the follow-up visit in the Method section as follows:

 Page 2, Lines 74-75 (2. Material and Methods, 2.1. Study Subjects)

“In addition, they received dietary counseling provided by registered dietitians for weight reduction and preventing weight regain at every visit.”

Page 12, Lines 348-350 (last paragraph of 4. Discussion)

“Since the single questionnaire was used in the present study, the results only assessed one aspect summarized in the questionnaire.”

We also responded to your comments point by point as follows:

#1-the title needs to be corrected. It is repeated twice.

Authors’ response

I am very sorry for the inconvenience. Since the authors described the term “(Change in Eating Behaviors Following Metabolic and Bariatric Surgery)” as a running title, it has been deleted in the revised manuscript.

#2-Figure 1: how many patients have been enrolled? 88 or 83? And how many patients underwent LSG? 57 or 55?

Authors’ response

I am very sorry. It was a simple mistake in the number of patients in the legend for Figure 1. Exactly, the number of patients who were enrolled in this study was 88, who underwent LSG was 57, and who were followed up after surgery was 49. Then, the legend for Figure 1 has been corrected. I greatly appreciate the reviewer for bringing this important error to our attention.

#3-Line 83: I think you could remove "as described previously" for what concerns Japan Society for the Study of Obesity. You explain it in the following lines.

Authors’ response

The authors totally agree with the reviewer’s comment. We have removed the term "as described previously" from this sentence. (Page 3, Lines 86-87)

Reviewer 2 Report

Manuscript that explores the factors that affect postoperative weight loss after bariatric and metabolic surgery, more specifically the changes in eating behaviors according to the Japanese Society for the Study of Obesity questionnaire.

Editing the text should consider:

-Table 1 is cut into two pages (4-5 /14) which makes it difficult to read

-The titles of tables 4 and 5 have two different font sizes.

- In the last line 353 (last conclusions) there is a sentence that should not be done: 2. Materials and methods

Author Response

Response to Reviewer #2

Reviewer comments for the author

Manuscript that explores the factors that affect postoperative weight loss after bariatric and metabolic surgery, more specifically the changes in eating behaviors according to the Japanese Society for the Study of Obesity questionnaire.

Editing the text should consider:

Authors’ response

We thank you for your time and effort in reviewing our manuscript. We really appreciate beneficial suggestions to improve the quality of our manuscript. We have responded to your comments point by point as follows:

#1-Table 1 is cut into two pages (4-5 /14) which makes it difficult to read.

Authors’ response

We totally agree with your comment. We have modified the size of Table 1 so that it does not cross pages. When editing, the authors will also ask the editorial office to keep Tables within one page.

#2-The titles of tables 4 and 5 have two different font sizes.

Authors’ response

I am very sorry for the inconvenience. As pointed out by the reviewer, the authors have corrected the font size to the same in the titles of Tables 4 and 5.

#3- In the last line 353 (last conclusions) there is a sentence that should not be done: 2. Materials and methods.

Authors’ response

In the originally submitted manuscript, there was no such term “2. Materials and methods”. As the reviewer pointed out, the authors have deleted this. (Line 361)
